# Psychological Impacts of COVID-19 on Healthcare Trainees and Perceptions towards a Digital Wellbeing Support Package

**DOI:** 10.3390/ijerph182010647

**Published:** 2021-10-11

**Authors:** Holly Blake, Ikra Mahmood, Gonxhe Dushi, Mehmet Yildirim, Elizabeth Gay

**Affiliations:** 1School of Health Sciences, University of Nottingham, Nottingham NH7 2HA, UK; ntxmy4@exmail.nottingham.ac.uk; 2NIHR Nottingham Biomedical Research Centre, Nottingham NG7 2UH, UK; 3School of Medicine, University of Nottingham, Nottingham NG7 2UH, UK; mzyim6@exmail.nottingham.ac.uk (I.M.); mzygd3@exmail.nottingham.ac.uk (G.D.); mzyemg@exmail.nottingham.ac.uk (E.G.)

**Keywords:** COVID‐19, pandemic, psychological wellbeing, digital, healthcare, students

## Abstract

We explore the impact of COVID-19 on the psychological wellbeing of healthcare trainees, and the perceived value of a digital support package to mitigate the psychological impacts of the pandemic (PoWerS Study). This mixed–methods study includes (i) exposure to a digital support package; (ii) participant survey to assess wellbeing, perceptions of work and intervention fidelity; (iii) semi–structured qualitative interviews. Interviews were digitally recorded and transcribed, data were handled and analysed using principles of thematic framework analysis. Participants are 42 health and medical trainees (9M, 33F) from 13 higher education institutions in the UK, studying during the COVID-19 pandemic. Survey findings showed high satisfaction with healthcare training (92.8%), but low wellbeing (61.9%), moderate to high perceived stressfulness of training (83.3%), and high presenteeism (50%). Qualitative interviews generated 3 over–arching themes, and 11 sub–themes. The pandemic has impacted negatively on emotional wellbeing of trainees, yet mental health is not well promoted in some disciplines, and provision of pastoral support is variable. Disruption to academic studies and placements has reduced perceived preparedness for future clinical practice. Regular check–ins, and wellbeing interventions will be essential to support the next generation health and care workforce, both in higher education and clinical settings. The digital support package was perceived to be accessible, comprehensive, and relevant to healthcare trainees, with high intervention fidelity. It is a useful tool to augment longer–term provision of psychological support for healthcare trainees, during and after the COVID-19 pandemic.

## 1. Introduction

The psychological impact of COVID-19 on healthcare workers is well documented [1,2]. A recent review and meta–analysis identified risk factors for psychological distress following the outbreak of any emerging virus [2]. Those at greatest risk tended to be female, younger, parents of dependent children, and those with an affected family member. Healthcare workers with pre–existing physical or mental health conditions, those who experience social isolation or prolonged quarantine, and those with concerns about infecting family members are more vulnerable to psychological impacts, as well as those who experience societal stigma from their healthcare role. Work–related risk factors for psychological distress include having greater contact with affected patients, having less clinical experience, and lacking access to appropriate organisational support and training, work wear or personal protective equipment (PPE). There are differences between clinical professions, with nurses found to be at greater risk than doctors in most studies [2].

Healthcare trainees share similar risk factors but are an under–researched and often overlooked group. They make a valuable contribution to the health and care workforce, and many senior trainees have been voluntarily mobilised into healthcare services, to support patient care during COVID-19 [3]. In general, college students have reported fear, disturbed sleep, depression and anxiety during COVID-19 [4,5,6,7,8,9,10] and anxiety levels for college students have been found to be higher than that reported by university staff [11] and almost double that of healthcare professionals [12]. The psychological impacts can be higher in students in their graduating year or those living in severely afflicted areas [5]. A study in the United States of America found that a high proportion of college students also reported concerns for the health of their families, difficulties in concentrating, decreased social interactions (due to physical distancing) and increasing concerns about their academic performance [7]. 

With regards to healthcare specifically, high numbers of healthcare trainees report feeling mentally unwell during the pandemic (e.g., [13], 52.4% of medical students) with delays in academic activities being positively associated with anxiety [14]. Many healthcare trainees report high ‘fear of COVID‐19’, particularly those who are female, younger, in their earlier years of training or with financial challenges [5,15]. In the United Kingdom (UK), the COVID-19 pandemic has resulted in significant impacts on clinical learning opportunities in health and medical education, and disruptions to clinical placements affecting students’ confidence and preparedness for clinical practice [16,17]. It is imperative that education systems adapt to meet the needs of healthcare learners during and beyond COVID-19 [18], and as such there has been a rapid transition to alternative forms of learning involving virtual learning, videoconferencing, social media and telemedicine [19], which has brought additional and unique stressors.

Undoubtedly, access to psychological support has been identified as important to mitigate the psychological impacts on healthcare workers of a public health crisis [2] and provide support during a pandemic for higher education students, globally [20]. Healthcare trainees should be afforded such provisions, which may help to improve aspects of their psychological wellbeing, for example, reducing fear and improving sleep difficulties [5]. Digital interventions can provide information, guidance, signposting and support while offering flexibility for use while remote working. At the time of writing, during the COVID-19 pandemic, universities in the UK have been largely delivering education remotely with plans to return to a hybrid model of remote and face–to–face delivery, post–pandemic. A digital support package is available that was released just three weeks after COVID-19 was declared a pandemic in the UK [21], with the aim of mitigating the psychological impact of COVID-19 on healthcare workers. The package was the first of its kind to provide education and supportive strategies focused on the psychological impact of COVID‐19, self–management approaches to self–care and psychological wellbeing. It is based on a conceptual model for mitigating the impacts of COVID-19 on health and care workers (Figure 1).

Content of the package draws on the principles of positive psychology which is advocated for the prevention of mental health problems (general population: [22,23]; student sample: [24]) and focuses on the strengths that enable individuals and communities to thrive. In the context of health and care workers’ wellbeing during a pandemic, this includes attention to the organisation (e.g., proactive organisational structures and approaches, communication strategies, prioritising staff wellbeing); leaders and teams (e.g., psychologically safe environments, compassionate leadership and role modelling, team collaboration, building team resilience, peer support); and individuals (e.g., building self–esteem and self–efficacy [25], self–care, staying connected and managing emotions). The content of the digital package is informed by effective health policy and leadership models [26,27,28], and aligns with the Five Ways to Wellbeing model [29,30] which identifies the five activities most likely to promote individual wellbeing: (i) connect (e.g. access social support), (ii) be active (e.g. self–care), (iii) take notice (e.g., risk awareness, mindfulness), (iv) keep learning (e.g., strategies for supportive teams, effective communication, cultural competence), (v) give (e.g., supporting others, psychological first aid). While the package is highly accessed globally, and has been found to be appropriate, meaningful and useful to health and care workers from diverse disciplines [31], the value to healthcare trainees is not yet established. The current study aimed to ascertain whether the digital package has relevance and value for healthcare trainees, as the next generation of the healthcare workforce. 

The aims of the research were to: 

(i) explore the experiences of healthcare trainees during the COVID-19 pandemic and any impacts on their studies and psychological wellbeing,

(ii) describe trainees’ mental wellbeing and perceptions of training (in terms of work stressfulness, satisfaction and engagement, presenteeism and intentions to leave).

(iii) determine the acceptability, fidelity and utility of a digital package to support psychological wellbeing in healthcare trainees,

(iv) establish recommendations for approaches to augment longer–term provision of psychological support for healthcare trainees, during and after the COVID-19 pandemic.

## 2. Methods

### 2.1. Study Design

This was a mixed–methods study involving individual qualitative interviews accompanied by a questionnaire survey with interview participants. The research was reviewed and approved on 11 June 2020 by the University of Nottingham Faculty of Medicine & Health Sciences Research Ethics Committee (FMHS REC 39–0620) and the study was pre–registered (PoWerS Study on clinicaltrials.gov, ID: NCT04429828).

### 2.2. Participants and Setting

Eligible participants were health and medical trainees registered for study at the time of the COVID-19 pandemic, purposively selected to represent diversity across higher education institution, gender and discipline of study (including medicine, nursing, and allied health). Female participants were purposely over–sampled. This was to reflect a higher proportion of women in the UK healthcare workforce overall (NHS employees: 77% female [32], and the gender balance in healthcare education (nursing students: 90% female; allied health students: 75% female [33]; medicine and dentistry students: 64% [34]. Participants were recruited from 13 universities in the UK, including the Universities of Aberdeen, Birmingham, Bradford, Cardiff, Central Lancashire, Coventry, Liverpool, Leicester, London (University of Central London, Imperial College London), Nottingham, Oxford Brookes and Teesside.

### 2.3. Procedure

Trainees were recruited over a six–week period between June and August 2020, via advertisements circulated by email and notifications on student–facing and healthcare social media sites. Interested individuals were asked to contact the research team to express their interest in taking part and were subsequently provided with an information sheet, a consent form and a link to an evidence–based digital package [21], described by Blake et al. [31]. This package was designed to provide psychological support to health and care workers during and after the COVID-19 pandemic. It covers psychological impacts of COVID‐19, psychologically supportive teams, communication, social support, self–care, managing emotions and further resources. Further details about the development of the package and fidelity testing with healthcare workers is published elsewhere [31]. 

#### 2.3.1. Data Collection

Data collection approaches are mapped to study aims and corresponding results sections (Figure 2). Data were collected by qualitative interviews and a structured survey was completed by all interview participants (to meet aims i–iii). Findings are synthesised in a discussion with recommendations (to meet aim iv). All data were collected by independent researchers who had no involvement in the design or development of the digital package.

(a) Qualitative Interviews 

Eligible participants were invited to take part in a semi–structured interview, conducted one–to–one by telephone or video–conferencing facility (Microsoft Teams) and audio–recorded with consent. Interested participants provided their contact details to a member of the research team and arranged a time that was mutually convenient. Participants were not reimbursed for their time, although to maximise participation they were offered the opportunity to take part in a prize draw for a £30 online shopping voucher. Consenting participants provided both verbal and written signed informed consent. Interviews were informed by a semi–structured topic guide (Appendix A) developed using the five–step process outlined by Kallio et al. [35]. The topic guide considered the following broad areas: to gather insight into the emotional highs and lows of being a healthcare trainee during the pandemic; to identify any facilitators, obstacles or barriers to accessing the e–package; to identify perceptions of healthcare trainees towards the value of the e–package during and after the COVID-19 pandemic; and to gather views on longer–term support for psychological wellbeing in healthcare trainees.

All project team members had completed training on interview approaches, research integrity, research ethics and Good Clinical Practice (GCP). Two researchers undertook interviews (IM, EG). Digital recordings of the interviews were then transcribed verbatim with 100% cross–checking for accuracy (conducted by IM, EG, GD). The number of participants interviewed was based on the number needed to achieve theoretical data saturation. With each interview conducted, the research team judged whether the data emerging was new and satisfied the research purpose. The researchers deemed no new data to emerge at the 42nd interview, at which point recruitment ceased. 

(b) Structured survey

Wellbeing and Perceptions of Training

Prior to the start of the interview, consenting participants were asked to complete a brief survey (Appendix A). This included questions about their age, gender, ethnicity, year of study, and whether or not they had worked in the UK health or social care services and specifically, in a COVID-19 high–risk area during the pandemic. The survey included measures of wellbeing, and perceptions of work adapted for a trainee sample, for whom work in this context is either study or clinical placements. 

The measures included a 14–item measure of wellbeing (WEMWBS: Warwick Edinburgh Wellbeing Scale, [36]). The WEMWBS is a widely used scale which is a measure of mental wellbeing focusing entirely on positive aspects of mental health. It has been validated in the general population, and student populations [37]. The scale has five response categories, summed to provide a single score, with higher scores indicating more positive wellbeing. 

The survey included the following four single–item global measures that were adapted for use with trainees (students). Job stressfulness [38] was measured by the item: ‘In general, how stressful do you find your course/training?’ with responses on a 5 point scale ranging from 1 = ‘not at all stressful’ through to 5 = ‘extremely stressful’. Job satisfaction [39] was measured by the item: ‘Taking everything into consideration, how do you feel about your course/training as a whole?’ with responses ranging from 1 = extremely dissatisfied through to 5 = extremely satisfied. Turnover intentions [40] were assessed using the item: ‘Are you considering leaving your course/training?’ (yes or no). Presenteeism was assessed using an item adapted from [41]: ‘As far as you can recall, has it happened over the previous 12 months that you have gone to work (including placement or studies) despite feeling that you really should have taken sick leave due to your state of health?’ with responses options 1 = no, never, 2 = yes, once, 3 = yes, 2 to 5 times, 4 = yes, more than 5 times. 

The trainees were also asked to complete the dedication sub–scale of the 9–item Utrecht Work Engagement Scale (3 items: DE2, DE3, DE4) [42] as applied to healthcare training. This required respondents to report their level of agreement with the following statements: ‘*I am enthusiastic about my training’, ‘My training inspires me’, ‘I am proud of the work I do’.* Responses were on a 6–point scale ranging from 0 (never) to 6 (always/every day). 

Assessment of Package Fidelity and Utility

In the same survey, intervention fidelity was determined through quantitative assessment of user experience, content relevance, utility and accessibility. We replicated procedures and success criteria described elsewhere for the evaluation of digital packages [43,44], using 20 questions about the usability and utility of the e–package with pre–defined success criteria that had been used in a prior evaluation of the same package with a sample of healthcare professionals [31]. 

#### 2.3.2. Data Analysis

Participant survey data were analysed using IBM SPSS Statistics for Windows, version 26 (IBM Corp., Armonk, NY, USA) [45]. Survey data were analysed by an independent researcher (MY) who had no involvement in the digital package development, or the recruitment, intervention, data collection or analysis of qualitative data. Descriptive statistics were provided by characteristics based on clinical exposure. Chi square test was applied to compare Likert scale items between participants who were working in a COVID-19 high-risk area, or not. Roughly normally distributed WEMWBS mean scores were compared between participants with COVID-19 high/low risk using independent samples *t*–test. 

Analysis of qualitative data was guided by principles of framework analysis [46,47] with a combined deductive–inductive approach. Framework analysis is a hierarchical, matrix–based method developed for applied or policy relevant qualitative research where timescales are limited, and the goals of the research are clearly defined at the outset. Interview data was mapped onto thematic matrices to allow for interrogation to address the research aims and objectives. Starting with a deductive approach, an analytic framework was used that pre–selected matrices to consider (amongst other things): personal wellbeing; impacts of being a healthcare trainee during COVID‐19; perceptions and views of the e–package; whether they had gained any new knowledge; what was the most and least useful aspect of the package; whether (and how) they had used any of the information or resources; and preparedness for return to the ‘new normal’ in the recovery phase of COVID‐19. Thematic summaries from the ‘personal wellbeing’ matrix established the level of user demand for the digital package during COVID‐19. Summaries from the ‘knowledge’ and ‘most/least useful’ matrices captured user–driven perspectives on the use of the digital package and how to improve educational provision around psychological wellbeing going forwards. The ‘*perceptions’* and ‘*impact’* matrices provided more general insight about how trainees felt that the digital package (and other educational resources or inputs) could support their psychological wellbeing during or after the pandemic, any potential influences of the support package on their wellbeing and work, and potential future sustainability of this digital support package. Then taking an inductive approach, we included additional themes generated from the data though open (unrestricted) coding. Higher–level codes within each theme were refined by grouping lower–level codes found in the data. One author (HB) generated the analytic framework and oversaw the process of analysis, with three team members (EG, IM, GD) populating the framework, interpreting data and validating the form and content of the framework. In the report of the findings, verbatim quotes with ID codes in parentheses representing gender, discipline and ethnicity have been used to represent each theme and subtheme.

## 3. Results

Forty-three people expressed an interest in participating, and 42 of these provided written consent, and completed both the interview and survey (see Table 1). The sample consisted of trainees in eight disciplines including medicine (Med: *n* = 28), nursing (Nurs: *n* = 7), midwifery (Mid: *n* = 2), physiotherapy (Phys: *n* = 1); ambulance (A: *n* = 1); other, e.g. health–related PhD (Oth: *n* = 3) and 19 (45%) were from Black, Asian or other minority ethnic groups (see Appendix A). The sample included participants at all stages of study (undergraduate, masters and doctoral level). Thirty–one per cent (*n* = 13) of the participants had worked in a COVID-19 high–risk area during the pandemic (e.g., dedicated COVID-19 +ve ward, intensive care unit, emergency department or ambulance services, ward with COVID-19 +ve patients, entrance meet and greet, staff or regular visitor to care or residential home, or other self–defined high–risk area). 

### 3.1. Survey: Wellbeing and Perceptions of Training

Scores on the WEMWBS ranged from 13–56 (mean = 36.1, s.d. = 8.74), with 61.9% of participants classed as having poor mental wellbeing. There was no significant difference in wellbeing scores among gender, year of study, discipline, or whether participants had clinical exposure during the COVID-19 pandemic or not. Among 42 participants, the majority reported moderate–to–high job stressfulness with relation to their course (83.3%, *n* = 35). Almost half of the trainees (47.6%, *n* = 20) reported presenteeism (going into placement/studies when they should really have taken sick leave due to their health), and one participant did this more than five times in the previous 12 months. Two trainees indicated that they had considered leaving their course, but neither reported any clinical exposure during the pandemic. The vast majority expressed that they were satisfied with their training (92.8%, *n* = 39). With regards work engagement, 79.6% (*n* = 33) reported that they were *enthusiastic* about their training (either often, very often or always), 74.8% (*n* = 31) felt that their training *inspired* them (either often, very often or always) and 79.6% (*n* = 33) felt *proud* of the work they did (either often, very often or always). There was no significant difference in overall work engagement scores with gender, year of study, discipline, or whether they had clinical exposure during the pandemic. However, trainees who had worked in clinical areas during the pandemic reported higher work enthusiasm compared to trainees who had not. Full details are available in Table 2.

### 3.2. Survey: Assessment of Package Fidelity and Utility

Results of the intervention fidelity testing (Table 3) show high fidelity and excellent implementation qualities. Twenty (of 21) pre–defined success criteria were met for the fidelity assessment (6/7 across delivery and engagement), and implementation qualities (14/14 across practicality, resource challenges, attitudes, acceptability and usability). While intervention receipt (perceived knowledge) rate appeared lower than the pre–defined figure of 90%, either immediate new knowledge enactment and/or intention to act on knowledge in the future was subsequently reported by 92% of respondents (39/42). Many healthcare trainees reported that following engagement with the package, they had already taken further actions (‘intervention enactment’) e.g., engaging with sleep and night shift tips, better communication with colleagues, emotionally supporting peers and family members, balancing responsibilities to themselves, family and friends, considering training in psychological first aid, engaging with advice around coping with emotions, and seeking individual discipline–specific help by accessing telephone helplines or web support found in the extra resources. Many had accessed the interactive elements (e.g., video clips), used apps signposted from within the package and shared the information with others. 

### 3.3. Qualitative Interviews

Interview data analysis generated three over–arching themes, with 11 sub–themes (see Table 4). Interview length ranged from 11 to 51 mins and the average duration of interview was 21 mins.

#### 3.3.1. Theme 1: Impact of COVID-19 on Studies

##### Sub–Theme 1: Level of Exposure to COVID‐19

The perceived level of exposure to COVID-19 through studies and clinical placements was highly variable in this sample. Many of the participants were deployed to frontline clinical areas during the pandemic or they were engaged in some form of clinical work outside of scheduled study time. This included work as a healthcare trainee, or bank staff (agency workers) in intensive care units, COVID (+ve) wards, the ambulance service, and care homes as well as other clinical areas perceived to be lower risk, such as outpatient clinics or non–COVID wards, without any positive cases. There was a general consensus that healthcare trainees had experienced greater impacts during the pandemic (both on their studies and through personal exposure to COVID‐19) than students from other disciplines. 

##### Sub–Theme 2: Impact of COVID-19 on Healthcare Studies

Participants alluded to the abruptness of the national lockdown and the significant disruption to student life: “*university just got cut short…when lockdown happened in March, I thought …my time as a student ended then… everything was just not involving university at all” (101FPhysioW).* The experience was described as “*daunting*” and “*no–one really knew what was happening*” *(121FMedM).*

There were mixed views about the transition to remote learning during the pandemic. Trainees recognised that the switch to online approaches to learning had been essential: “*that’s the cards we've been dealt, and obviously I don’t really think there’s any, there’s no other way of doing it*” *(109FMedW).* For some, the change of study environment was perceived to negatively impact their productivity and engagement: “*I personally haven’t been very interactive with some of the stuff, because I find it quite difficult to learn like online through that kind of method*” *(127FMedM).*

There was a perception that academic staff understood the challenges of remote working for students and had quickly adapted their approach and lecture materials to accommodate this: “*I feel like the lectures are more informative now, because the lecturers are working harder, because they know that we can’t ask them questions as we can in a, you know, in–person lecture*” *(108FOthM)*.

The changes to timetables, cancelled lectures and training sessions had resulted in fewer support sessions for some participants (e.g. revision sessions for exams), and areas of learning that trainees perceived had not been covered adequately, or at all: *“we lost out on a lot of like communication sessions, they were never picked back up” (112FMedW)*. Nevertheless, participants spoke of adjusting to remote learning over time, and highlighted the availability of online lectures and learning materials not only from their own institution but from other universities and healthcare organisations, demonstrating a marked increase in the visibility of shared learning resources as a result of the pandemic. For some, the transition to online learning approaches had increased their capacity for independent study and actively seeking out learning materials by necessity: *“I’ve had to do more, more so by my own initiative by joining online webinars and you know, finding stuff out by myself instead of it being provided for me” (124MMedW).*

Some expressed a preference for (and reminisced about) face–to–face learning, in particular since this was seen to be a more effective platform for questions and discussion. A minority of participants perceived greater challenges in accessing individual–level support from academic staff during this time. The perceived inability to interrupt online sessions mid–flow with questions made it difficult for some trainees to engage in complex discussions via these remote methods. This seemed to be more problematic for trainees learning in large cohorts, for example, where teaching delivery was in the format of large–group online lecture. In those healthcare disciplines where cohorts were smaller, it had been easier to sustain small–group tutorials which had led to high quality discussion and feedback and a willingness to ask questions. 

The rapidity of the transition to online learning and adapting to new approaches and systems had generated significant stress for many participants: “*it was all a bit stressful because I’ve never done online studying before*” *(122FMedM).* This was notable for those participants who had outstanding exams or assessments: “*I felt like we weren’t getting the right education we should’ve been getting, and our exams got cancelled, everything got crammed into one which was really stressful” (122FMedM);* “*I feel like we weren’t assessed properly and so I hadn’t studied properly because everything just seemed a bit…no–one really knew what was going on with our exams and stuff*” *(108FOthM).* For some of the trainees, the pandemic had led to a dramatic shift in assessment timings which was generally perceived to have been inevitable. Some of the participants admitted that the cancellation of scheduled lectures and placements had offered some respite from the intensity of healthcare training: “*a lot of people were really burnt out at that point in March, very anxious about exams, so when they cancelled everything, it was actually quite a relief*” *(111FMedW).* However, there was a certain level of anxiety regarding the additional workload that would be added to the next academic year. The emotional burden of the impacts of COVID-19 on workload and timings was evident, and one trainee described this as: “*emotionally taxing, stressful, because it’s all being like crammed into a short period of time, it might be quite stressful to, to get it all done*” *(111FMedW).*

Balancing academic studies with personal responsibilities during the pandemic generated stress for some. Assessment deadlines were automatically extended for students at some institutions, and this was appreciated, for others this was not the case: “*I’ve been having to do all this and then like neighbours’ shopping and you know a bit of, a bit more time to study would’ve been appreciated but they didn’t extend any deadlines” (140FNursW)*.

There were marked differences in the experiences of trainees at different stages of their study, with greater impacts on clinical experiences perceived by those in the later stages of study and those who were deployed into clinical areas during the pandemic, and fewer impacts perceived by those who were in their earliest stages of study with fewer placements, or for those who had already completed their placements, taught sessions and assessments prior to the lockdown. Those who had experienced disruption to clinical placements had concerns about their development of clinical skills, and overall quality of their education. For some, the disruption had significantly reduced opportunities for clinical experience. One trainee reported: “*The ambulance service won’t take us…we weren’t allowed on our second placement block which was supposed to start in May. They won’t take us until this thing has blown over*” *(103MParaW).* A medical trainee stated: “*We should have been now just doing two years straight of placement, but it’s been put on hold*”; “*we’ve missed all of that and had to kind of learn something that’s very physical and very practical in a non–physical and non–practical way*” *(110FMedW).* Some participants perceived that clinical exposure had been prioritised for those in later years of study (e.g. the final years of medical training) and while they were not critical of this approach and recognised the need for certain groups of students to attain the correct number of hours in practice to qualify, it left some trainees in the earlier stages of study concerned about the lack of opportunities during this time and any long–term implications of this: “*We’re tryna learn things that can’t be learnt from home at home basically and its very, very, up in the air so we don’t really know when we’re gonna be back on placement, and there’s obviously a lot of anxiety around how it’s gonna impact us now cos we are very behind in terms of where we should be at, at this point*” *(110FMedW).*

There were concerns raised about clinical subjects being taught fully online, and this negatively impacting their learning and the development of clinical skills: “*there’s only so much you can learn you know from like a Teams lecture like you do need like the hands–on practice you know on the wards*” *(112FMedW).* However, the efforts of clinical teams to develop new learning approaches for trainees was welcomed: “*Making us take histories from each other via Microsoft teams and …for … critical care where you’re kind of trying to hand back, handover patients via Microsoft teams to just kind of try and put some interactivity into it*” *(113MMedM).*

For those in their final year of studies, the transition from trainee to healthcare practitioner was perceived to have accelerated. For some, the pandemic seemed to have shifted their identity as a ‘student’ to that of a ‘professional’, although some admitted to struggling with the adaptation to their new role: “*still technically finishing my final year… with dissertation and things…and then having to just go straight into practice as a qualified member of staff, it is really tough” (101FPhysioW).* Some students raised concerns over the impact of the pandemic on specific processes, such as the application process for foundation year doctors (in the UK this is a two–year, general postgraduate medical training programme which forms the bridge between medical school and specialist/general practice training) or the completion of elective placements (in the UK, the content and setting of elective placements are largely decided by the student undertaking it). Those participants who were on clinical academic pathways (a planned progressive development through undergraduate, masters, doctoral and post–doctoral levels of study, spanning clinical and academic environments) reported significant challenges of engaging in part–time study alongside their clinical roles, particularly those who had been redeployed during the pandemic to intensive care or COVID-19 wards, leaving little time for their studies. 

Some students who were not working in clinical environments during the pandemic reported feeling ‘cut–off’ from the university during the early months of the pandemic. Feelings of isolation were generally reported by students who were registered for part–time study, or those who were undertaking doctoral degree programmes in the healthcare disciplines, rather than undergraduates who were part of a large cohort of trainees, or those working in clinical environments. 

The need for regular communication was paramount. Trainees generally reported a high level of support from their institutions. They applauded the detailed information that had been delivered at speed to assist them with decision–making around opting in for COVID-19 extended placements to support frontline healthcare staff. The majority were positive about the regularity and clarity of information and communications coming from their university at the outset of the pandemic: “*We had so much information on that, meetings by zoom and emails, constant communications so I felt really supported to make my choice to opt in for that placement*” *(102FNursW).* However, some of the trainees had a less positive view of communications and it was apparent that the frequency, and clarity of information was not consistent across institutions. Some trainees felt that their university had not efficiently communicated changes in working hours and expectations of trainees to placement mentors, which caused difficulties on clinical placements and was anxiety–provoking.

While there appeared to be a satisfaction with the level of information provided around the move to remote learning, or the processes for opting in for clinical placements, some trainees felt there was a lack of detail in areas that mattered to individuals personally. For example, one medical trainee referred to COVID-19 risk assessments being conducted on healthcare students with medical conditions, in preparation for practice, but they perceived there to be a lack of clarity around how this related to their personal circumstances, particularly if they lived with a vulnerable person: “*what if you live at home with, you know, who, someone whose severely immunocompromised or something like that, how do you go around that? Are you expected to move out of that home?*” *(110FMedW)*

The uncertainty of COVID-19 had led to confusion that was seen to impact on the clarity of communications from the university to trainees across the board. Trainees viewed late messaging from course leads and changing plans to be a significant source of stress which prevented planning, and adequate preparation related to their studies or placements: “*here’s the plan, and then a week later it changes and then it might go back to the original plan, and so it's a bit hit and miss, because you can't really adequately prepare*” *(127FMedM).* However, trainees commonly made a point of stating that the situation was outside the control of their educational institution: “*I don’t blame the medical school whatsoever, but nobody’s really knowing the future, what's going to happen*” *(127FMedM).*

Trainees often referred to being appreciative of the support they had received from clinical educators, academics, and from their university more broadly. They discussed the way in which support had been rapidly mobilised by academic teams: “*I think they've [the medical school] kind of done the best–case scenario in a worst–case situation*” *(109FMedW).* Participants reinforced that their universities had endeavoured to provide them with opportunities, whilst ensuring no trainee was exposed to situations that they would be ill–equipped to manage: “*as a student it’s very flexible … we’ve very much been told that if there is any scenario that we don't feel we’re comfortable with in terms of COVID, then we are allowed to opt out of it*” *(109FMedW).*

Building high quality relationships with academic tutors and feeling able to access support was seen to be essential for coping during the pandemic. One trainee discussed the benefits of receiving regular contact, one–to–one explanations and support from a tutor: “*[she] really made me understand what is going on and how to overcome these challenges, and basically I could say that it helps to prepare yourself and be ready for any hard time*” *(106MNursM).*

#### 3.3.2. Theme 2: Emotional Impacts of COVID‐19

##### Sub–Theme 1: Emotional Highs of the Pandemic

Despite the stress of the pandemic, among trainees that had been deployed to support the UK National Health Service (NHS) during the crisis, some viewed this moment as their ‘calling’: “*very excited to want to be on the front line, to support and do my duty. That’s why I trained to be a nurse*” *(102FNursW)* and an opportunity to: “*witness the changing of the system*” *(102FNursW).* There was a prevailing sense of pride in viewing themselves as the next generation of healthcare staff: “*I feel like I am really enjoying working and enjoying being part of it and seeing all of this change…we’re kind of that next generation of staff*” *(101FPhysW).* One nursing trainee described the opportunity to work alongside patients who have suffered with or been impacted by COVID-19 as a ‘privilege’. The learning opportunities provided by the pandemic were seen to be exceptional; observing major organisational changes such as areas being rapidly segregated for COVID cases, and the implementation of national and local policy changes in response to a pandemic. The value of teamwork in healthcare was very much at the forefront: “*I’ve been able to be part of team who have survived the COVID outbreak and still stuck together, and I think that’s brought a sense of unity*”. On the whole, trainees demonstrated that there had been a wealth of learning opportunities that had arisen from the pandemic. Many of them found value in the opportunities to see new ways of working, and work with new teams in different areas. They spoke of huge collaborative efforts to troubleshoot, find solutions and new approaches during the crisis with a shared goal of providing high quality patient care. 

The trainees felt they were able to make a positive contribution to healthcare when it was critically needed, and in very diverse ways. Those who were unable to independently provide clinical care reported having contributed to the pandemic through infection prevention and control action (e.g., cleaning of ward areas and emergency vehicles), or supporting patients: “*When it comes to working in hospital, it would be going and sort of either feeding or speaking to patients who are lonely because they can’t have their visitors. Or it may be going and helping HCAs [healthcare assistants] with their daily tasks working on COVID and non–COVID wards*” *(126MMedM).* While not all trainees were able to offer their support in this way, for those that could, these actions appeared to engender a sense of personal value, and their contributions were often confidence building.

Despite the stress experienced by the majority of trainees, those that had worked in high–risk areas during the pandemic, such as critical care, COVID-19 positive wards, or care home facilities perceived a positive side to the situation: “*I can tell a story that actually I did my last year of nursing during a pandemic, that’s a story and an achievement in itself*” *(102FNursW).*

##### Sub–Theme 2: Emotional Lows of the Pandemic

The stress and anxieties surrounding the impact of COVID-19 on studies was evident across the sample. For some of the trainees, the lockdown had taken a significant toll on their mental health: *“Staying at home really affected my mental health because towards the start of it, of lockdown, I was a bit, it got me depressed because I couldn’t go out anymore and nothing was the same, so I was just stay in bed all day, do nothing and barely even feed myself and things like that” (121FMedM).*

Some of the participants reported a lack of recognition of the impact of the pandemic on their mental health, and experienced feelings of abandonment by their academic institutions: “*I felt quite let down with what the university provided us with in terms of support and help for our well–being during this time*” *(101FPhysW).*

Course–related stress and anxiety primarily related to the confusion surrounding processes, cancellation of assessments, rapid changes in traditional learning approaches and/or the reduction or cessation of clinical placements. There were high levels of anxiety among those who were working in clinical environments during the pandemic; there were initial concerns about access to appropriate personal protective equipment (PPE), and access to training in appropriate use of the equipment, although these concerns reduced over time with increasing availability of PPE within the NHS. For some, the extent of the changes, and continual high stress had been overwhelming and was showing no sign of reducing: “*I’ve been working since April and even now I’m still feeling very overwhelmed*” *(101FPhysW).*

Participants discussed feelings of helplessness when faced with patients who were seriously ill with COVID‐19, and the uncertainty of not knowing how ill a patient would become presenting a mental challenge. Trainees spoke of learning how to step back in cases where there was nothing further they could do: “*seeing those really poorly people and actually not been able to do huge amount for them and kind of just letting things happen…it’s been really hard to do and kind of wanna get stuck in, actually most the time the having to, urm really just kind of, fight it for themselves as well, with our help…*” *(101FPhysW).*

The impacts on trainees of what they saw, and heard, from healthcare professionals during the pandemic was at times profound, but trainees had found the positives by using reflection to channel negative experiences and emotions into learning experiences: “*the most shocking thing for me was just hearing the staff’s experiences where they were frontline in the, what we call ‘hot Covid’ areas. Hearing their stories, that was quite impacting, and I actually did a reflective piece on that as part of my work and the moral distress they experienced during the pandemic” (102FNursW).*

Participants expressed a fear over the possibility of a second wave, and the impacts that might have for them as healthcare trainees, with relation to how they would cope. Those in their final year raised concerns over whether it would hinder their transition to becoming a healthcare professional. 

Trainees commonly experienced conflicting emotions due to perceiving themselves (and others perceiving them to be) part of the healthcare workforce, yet at the same time, limited in knowledge and experience by nature of being a trainee: “*it’s quite tough like being a student, because you feel like you should be helping, but, we’ve, we haven’t really been taught that much, like, clinical stuff yet in the preclinical stages. So, I feel like if we did help out, we wouldn’t be able to do a great deal*” *(107FMedM).* Similarly, there were feelings of ‘imposter syndrome’ expressed, with stories of discomfort at times when trainees perceived themselves to have limited knowledge but were seen by those in external communities as medical experts. Due to the national uncertainties around the transmission of COVID-19 and expectation behaviours, many trainees had been asked for advice by friends and family members which at times, had been a source of stress: “*it’s difficult to try and, try and help reassure people, without being worried about giving them, not necessarily false hope but … false information*” *(125MMedW).* Comparable discomfort was experienced by some on their clinical placements: “*there’s been more expected of you than what you would’ve liked, and you’re sort of assumed to know things that you don’t know*” (139FNursW).

Trainees were concerned about their own exposure to COVID-19 when working in clinical environments, and the risks to children or other family members with chronic conditions which increased their anxiety. This had led to some trainees making significant personal sacrifices in order to attend clinical placements, such as sending their children away to be looked after by relatives during this time.

Personal circumstances generated varying levels of stress—those living with their families mostly reported stress and anxiety associated with COVID-19 transmission, or caregiving responsibilities, and isolation due to not living with, or having face–to–face contacts with their peers. For those living in university accommodation or shared housing off campus, living with peers during the pandemic was at times supportive, and at other times a more challenging experience: “*living in a group together in lockdown was…quite a new experience, it had its ups and downs sort of took a toll on everyone’s mental health” (139FNursW).*

The psychological impacts of the pandemic were not always immediately evident to some trainees while working in clinical environments who, aside from those in intensive care units or dedicated COVID-19 wards, tended to view themselves and colleagues to be engaging in “*business as usual*”. However, several of the trainees alluded to experiencing delayed shock and psychological impacts once their shift was over: “*After we would get home, home into the bays, we would leave the ambulance, put a bit of masking tape over the doors, and say it’s a Corona–Van, and er, get another ambulance….it started to make an impact sort of, when you had three ambulances sitting in the bays, that you’re not allowed to touch*” *(103MParaW).*

##### Sub–Theme 3: Discipline–Specific Impacts

Trainees often referred to the uncertainties in how to manage a patient with COVID-19 and worries about adverse effects of any interventions or treatments they might provide. This was particularly evident in comments made by allied health professionals: “*… as a physiotherapist, going in, providing treatment and thinking okay I’m not sure how that’s going to go. Erm, whereas with kind of some other diseases and illnesses that we know a little bit more about, you can be a bit more certain of what your effect of your intervention’s gonna be” (101FPhysW).*

Moral distress was a common theme among nursing trainees, for whom education focuses around the delivery of person–centred care. Some participants reported having struggled with the concept of having to prioritise patient safety over holistic patient care due to the excessive demands on healthcare systems and resources: “*it got to a point where some nurses were looking after four critical care patients, rather than one or two. So, as you can imagine the workload had increased, so they often felt that they couldn’t fulfil their nursing duties, not fully as they would pre–covid, and things like mouth care was being missed … because there was no time” (102FNursW).*

Medical trainees experienced a certain level of pressure, mostly from healthcare professionals in other disciplines, who seemed to have high expectations of them with regards level of knowledge and understanding of virus transmission: “*because I’m a medical student it was a bit more stressful for me because people expected me to know something about this virus” (122FMedM).* Medical students were more likely than students from other disciplines to report feeling unable to disclose concerns relating to mental wellbeing, or worries about the future, with peers, academic staff, or clinical mentors.

##### Sub–Theme 4: Ethnicity–Specific Impacts

Those trainees who were from minority ethnic groups, expressed a high level of fear and anxiety around COVID‐19. These students expressed concerns about their exposure to COVID-19 on clinical placements, and the implications that may have for vulnerable family members. 


*“I’m actually going to bring the coronavirus home…infect those who are, you know, immunocompromised and who are more likely to … suffer from long lasting damages from the coronavirus or…possibly even death”*

*(108FOthM)*


One nurse trainee spoke of her competing responsibilities in the dual role of parent of a child from an ethnic minority, as well as a healthcare worker and the life decisions this had led to: “*for me as both a healthcare professional in training and as a mother, seeing that COVID impacts on certain communities, it doesn’t discriminate as such, but I think it does favour the black and minority ethnic groups” (102FNursW, mother of dual-heritage child).*

Those trainees who identified as White recognised the increased risks for ethnic minority populations primarily from the media coverage, and believed they had a certain level of social responsibility towards patients from higher-risk groups:


*“…it’s changed my mind, it’s changed my opinion in terms of you just got [COVID‐19], for me I’ve got to look out even more so for … those types of patients that I really am looking towards their best interests and making a big conscious effort to really ensure their recovery from this illness“*

*(101FPhysW).*


Trainees who had accessed COVID-19 wards spoke of the impact of the virus on certain populations, and the visibility of this in certain hospital wards: *“I was asked to do a few shifts on the COVID areas, so I wore the full PPE, and when I walked into there, the patients were majority of black and minority ethnic groups, and that was like whoa. And that took me back, as I live in a community where there is a lot of black and minority ethnic groups…so I was like woah actually this is very close to home, and that got me worried for their health, for their safety”.*

##### Sub–Theme 5: Return to the ‘New Normal’

Concerns about the future were primarily related to an uncertainty around where trainees would ‘fit’ in the new normal of healthcare, the impacts of the pandemic on healthcare education and whether gaps in provisions would reduce preparedness for practice. There was also a recognition that the psychological impacts of the pandemic would generate support needs that may continue for the longer term. Healthcare trainees recognised that they would need to adjust to studying and working in a new way and there was widespread concern about what the ‘new normal’ post–COVID would entail, particularly relating to practice placements. It was generally accepted that most trainees would return to practice ‘*before we are post–pandemic*’ and all participants were anticipating change to their placement experiences as a result of this, associated with the strain on healthcare services during this time and the high workloads of healthcare professionals. For some, concerns about their exposure to COVID-19 patients, and a lack of clarity around what might be expected of them on future placements generated significant anxiety: *“I think it’s quite scary…we’re not being thrown into it by any means, but it is a bit fearful … you don't really know what you’re going into.” (127FMedM)*

Some of the trainees were worried about the level of support that would be provided from clinical mentors and supervisors going forwards because of competing demands on their time, and this led to trainees feeling concerned about the potential for gaps in their knowledge that could impact on patient care in the future: *“A lot of us are feeling like we’re gonna be a burden on the healthcare staff ‘cause they’re all gonna be so busy, so a lot of us do feel a bit apprehensive about, you know, asking questions, asking to be involved” (112FMedW); “there may be a gap in knowledge … for certain things which were rushed or perhaps not given enough attention to” (138MMedM).*

While some trainees enthused about returning to face–to–face teaching and placements and spoke of feeling ‘*energised**’* from time out: *“I’ve had time to refresh” (110FMedW)*, it was recognised that others would need support to make this transition. There was a prevailing view that living through the COVID-19 pandemic, and the significant disruption to healthcare education during this time would be likely to have had long–lasting psychological impacts on many healthcare trainees. Participants commonly alluded to their own mental health concerns, or those of others, and proposed that mental health problems were likely to become more prevalent in healthcare trainees over time. 


*“I don’t think it will be normal. Everything will change. I don’t think people will be the same as before, especially me, talking personally. I’ve been impacted by this honestly, mentally, emotionally.”*

*(106MNursM)*



*“I think it [low mood] will decline quite significantly, I think a lot of people have been just quite low mood anyway purely because of the lack of social interaction and the fact that we’ve now been inclined to revise for three and a half months for medical exams.”*

*(129MMeMB)*


Wellbeing support for trainees was not perceived to be universally available. Healthcare trainees alluded to significant differences in the value placed on student wellbeing by different institutions and between disciplines: *“…the university that I was affiliated with, I didn’t hear of any sort of support systems from them*” *(132FOthW).* Further, trainees perceived there to be notable differences between the level of concern for welfare shown by individual staff members (e.g., supervisors, tutors, mentors). Some trainees reported a high level of support from staff, whereas others experienced challenges in accessing support or a lack of compassion and understanding around issues of wellbeing or personal circumstances impacted by COVID‐19. The support of fellow trainees and friends was seen to be a mitigating factor against psychological stress, and trainees with robust support systems felt more confident in their capacity to cope with returning to studies and placements than those with fewer support networks.

Participants discussed possible approaches to supporting the ‘return to the new normal’. Validating mental health concerns was seen to be essential, and trainees highlighted the value of this validation coming from senior staff (‘*at the top, reassuring you’ (127FMedM)*) as well as tutors. A widely shared sentiment concerned the ‘normalisation’ of psychological impacts of COVID‐19, since healthcare trainees in particular tend to suppress mental health concerns and struggles due to the nature of their degree: *“I think a big thing within the NHS is that people don’t feel like they can talk about something and they don’t feel like they’re almost allowed to have any sort of mental health or any sort of detriment to them because they’re worried that that will put them and their job at risk” (142FMedW)*.

It was universally accepted that trainee stress levels could be reduced through more effective communication from the university throughout decision–making processes rather than waiting for top–down delivery of final decisions; *“Be more open about what’s happening I think behind the scenes.” (122FMedM).*

There was a strong view expressed that wellbeing should be embedded within education and training programmes for healthcare trainees, with efforts to build robust relationships and support systems within cohorts, and timetabled opportunities for wellbeing support “*instead of just relying on students seeking out help” (120FMidM).* Many of the trainees advocated that more mental health awareness was needed generally within universities to reduce the stigma associated with mental ill–health. One participant suggested social media might be one approach to reaching healthcare trainees with mental wellbeing campaigns. There was a general consensus that universities (and workplaces) needed to reduce the waiting times for counselling services in order for them to be timely in preventing or managing the likely escalation of mental health concerns post–pandemic. It was also advocated that trainees would benefit from regular opportunities to openly discuss their worries or pastoral concerns with either a member of staff or another trainee in ‘*quick, one–to–one meetings, just to check in’ (112FMedW).* Active listening, understanding and compassion were seen to be critical in helping trainees to successfully navigate through this difficult period.


*“If supervisors can understand us, that we are humans, you know we have got feelings, we feel, you know what I’m saying. That is what I think is most important during this time”*

*(106MNursM)*



*“It’s been quite difficult to find who to talk to…if everyone is more supportive, ready to listen and actually willing to listen and makes the time for that”*

*(101FPhysW)*


#### 3.3.3. Theme 3: Digital Support for Psychological Wellbeing (the E–Package)

##### Sub–Theme 1: Usability and Engagement

The vast majority of participants felt that the package was easy to access and navigate, including those who did not consider themselves skilled with information technology: “*no one said that they’ve struggled with it at all*” *(110FPhysW).* There were no reports of technical issues, and the package was viewed to have high functionality. The layout was deemed to be appropriate and facilitated engagement with the materials: “*I really liked just how simplistic the layout was and how concise it was*” *(127FMedM),* “*the way it was laid out was brilliant with the sections, so you could jump to the section if you needed to*” *(110FNursW).* Whereas some students found that the package was long, others referred to the value of subsections for selecting relevant content: “*there’s a lot of reading to it but they also seem to be sort of quite short bursts of it which makes it doable and obtainable, and it’s clearly set out so you know you can read the bits that apply to you*” *(140FNursW).* The collation of materials into a single package was seen to increase the accessibility of support for psychological wellbeing. The content was seen to be relevant to healthcare professionals and students alike, and the package was seen to be a ‘go–to’ resource for many: “*it’s kind of become like a, like a small encyclopaedia” (116FOthW).* Participants valued the variety in terms of the media used (text, audio podcast, video) and presentation of materials. Overall, the content was unanimously seen to be highly comprehensive and appropriate for healthcare trainees of any discipline “*it goes into a good level of depth, and then if they want to find out more about it, there were some quite good signposts” (107FMedM), “I like how it has so many links to lots of different resources that you’d otherwise have no idea existed” (136FMedM).*

##### Sub–Theme 2: Areas of Learning and Impact

Overall, the vast majority of healthcare trainees reported that they learnt something new from the package. The section on managing emotions was deemed to present the most important content by most of the trainees. There was a general consensus that one of the most important messages trainees took away from the package was the importance of having a self–awareness of their psychological wellbeing and reflecting on their own emotional wellbeing: *“managing the emotions and understanding the emotions, like the anxiety and the anticipation. That section was really, really good.” (104FNursW);* “*it’s actually just making myself aware of feelings that I might have maybe suppressed because of the psychological impact of them.” (101FPhysW)*

Participants admitted that topics such as stress, anxiety and low mood were not typically discussed in their areas of study or professions despite the fact that mental health was something many healthcare trainees struggled with: *“you just have to deal with it’ (122FMedM).* Several of the participants discussed a newfound recognition of the signs and symptoms of stress and alluded to feeling sanctioned by the package to consider their own wellbeing: *“it helped me spot…warning signs of being really stressed…the feeling of being stressed and panicked, I should probably get help for that” (123FMedM).*

There was a recognition that burnout was common in healthcare professionals and healthcare trainees, although the participants reported that they had little prior knowledge on this topic: *“so it talks about burnout … I didn’t really know like the signs …I didn’t really know about the different options before the package.” (107FMedM).* The inclusion of material on moral injury and guilt was valued. Many trainees had experienced feelings of guilt but felt unable to discuss this openly: “*I don’t think it’s something spoken about a lot… in a time when so many people are passing away and getting ill, you’re always gonna tell yourself, oh I should have done better I should have done better, and I love the fact that I think a recurring theme was you are doing enough.” (127FMedM)*

Healthcare trainees spoke of the value of learning about the psychological impact of COVID-19 in different types of job role, and this engendered a sense of social responsibility towards their peers and colleagues in health and social care: *“Now I know sort of which friends perhaps I ought to check in on and see how they’re doing and how they’re coping with this.” (142FMedW).* It was particularly notable that none of the participants had undertaken any training relating to mental health (either as part of their course, or elsewhere) and many highlighted a lack of knowledge and low confidence around signposting people with mental health concerns. As such, the inclusion of material on Psychological First Aid within the digital package, was welcomed. 

Participants valued the inclusion of practical advice related to work breaks, and self–care. Most of the healthcare trainees talked about difficulties with disrupted sleep during the pandemic and the challenges of getting into sleep routines. Practical advice related to sleep and managing fatigue was therefore seen to be particularly useful and something that healthcare trainees could put directly put into practice after accessing the package, particularly those who were on clinical placements and working night shifts: *“it was the information about kind of sleep …managing fatigue, which was very useful” (113MMedM), “I really liked how there’s like focus on sleep a lot because … I think that’s such a big thing” (123FMedM).*

The collation of resources for wellbeing within a single package was highly valued. Many trainees reported that one of the most important aspects of the package was the fact that it signposted them to resources such as apps and links related to wellbeing which they had previously not been aware of and which they could now access when needed: *“like mindfulness and managing stress, anxiety and low mood and having it all there and it’s so accessible for everyone because you’ve got all the links in one place.” (128FMidW); “I didn’t know that the NHS…has free access until December to a load of wellbeing apps.” (103MParaW); “there is a lot of…help out there but people don’t know where to go to get it so if this tool was more widely available, people would actually use these resources because they are so needed.” (134FMedB)*

For some, this provided a resource that would help them with signposting when discussion psychological wellbeing with the peers and colleagues: *“if they come in and have a chat to me, I’ll be like right, you know, go and have a look at this” (118MMedW)*


Materials on leadership were seen to be more or less useful to trainees depending on the stage of training the individual was at, with those working on clinical placements or associated with clinical teams being more able to set the learning in context than those in their first year of study. However, several participants reported that the management and leadership section helped them to understand the context of leadership during a pandemic, and more broadly, what might be expected of seniors and mentors when trainees entered clinical practice: *“if you’re…a leader, what to do if some of your colleagues and staff are affected by what's going on … as a leader …it’s important to be aware of how your staff are feeling as well as the job they’re doing”. (107FMedM); “I think about what our employers…have to go and implement, or what your managers should implement” (126MMedM)*

Understanding stigma related to COVID‐19, and communication about COVID-19 was an important area of learning for these healthcare trainees. Participants reported that stigmatising language was an issue they had not previously considered, but learning from the package would influence their future communications with colleagues, their patients, and the general public: *“it’s quite easy for us to go…assuming that we understand what communication is, but actually we’re working with people who don't have the same experiences or backgrounds as us. So, it’s really clarifying what communication is expected from us with everyone, not just our patients.” (126MMedM); “because I was guilty of using those words, you know, automatically so it will, like, change my conversations now”. (112FMedW)*

Many of the participants alluded to an increased sense of cultural competence following engagement with the package. For example, trainees spoke about new learning related to Ramadan and the impact of fasting on patients and shift–working colleagues: *“it wasn’t something that I hadn’t necessarily immediately thought of” (110FMedW)*

##### Sub–Theme 3: Areas for Future Development

Although trainees were positive towards the package accessibility, function and content, several of the participants commented on the length of the package. It was proposed that a shorter package, covering few topic areas perhaps in greater depth, may be valued by healthcare trainees to avoid ‘*information overload*’ in a single package. A potential solution was to segment content into a series of shorter packages to create a collection of tools each covering one specific area. There were no technical problems reported. Despite the function to allow return to the main menu, it was noted that it was not possible to go backwards using an arrow key, and for a minority of users, it was unclear how to return to the main contents page. It was proposed that the functionality of the package could be improved to make it more accessible on a mobile device as many trainees had chosen to access it on their phones. Some participants proposed suggestions to increase the accessibility of the package. For example, one participant suggested colour–coding sections, and another proposed the use of audio subtitles to be more inclusive to people with disabilities. One participant proposed that the package could incorporate more video clips to increase interactivity.

Whilst views on the relevance of each section varied according to participants’ prior knowledge and experience, trainees were broadly positive about the utility of the information provided to themselves, or others. The package contained student–specific information in the additional resources section, but the main body of the package was targeted to healthcare professionals. Trainees found the package useful but expressed a desire for additional content that was specific to students (noting that some was available within additional resources), and specific to their own educational institution. One example was the inclusion of strategies and tips for coping on placements, particularly those that were some distance from their homes and required stays in hospital accommodation. 

## 4. Discussion

This is the first study to explore the perceived value to healthcare trainees of a digital intervention designed to mitigate the psychological impact of the COVID-19 pandemic on health and care workers. 

First, this study contributes to an emerging evidence–based on the impact of the COVID-19 pandemic on healthcare trainees. Our sample was diverse; participants from 13 universities, studying medicine, nursing or allied health subjects, at various stages of training, shared their experiences. Irrespective of level of exposure to COVID‐19, most of our participants reported high levels of job stressfulness associated with healthcare training, and low mental wellbeing was evident in almost two thirds of our participants. This aligns with prior research identifying high stress, anxiety and depression in healthcare trainees, pre–pandemic (e.g., health professions: [48], medicine: [49,50,51], nursing: [52,53]. The COVID-19 pandemic is associated with high levels of psychological distress, globally [9,54,55], particularly in healthcare workers [56,57,58,59,60], younger populations [54,61,62] and student groups [4,6,7,8,9,10,15,63], including healthcare trainees [5,13]. 

Our findings support prior research and suggest that mitigating the impacts of COVID-19 on mental health of healthcare trainees should be prioritised by higher education institutions and healthcare employers. However, current provisions to support mental health appear to be inconsistent across health and medical disciplines, and institutions. There is a perceived stigma surrounding both prevention and help–seeking for mental health concerns, particularly among medical trainees. Efforts need to be made to validate those with mental health concerns by normalising discussion about mental health and promoting interventions to support healthcare trainees with all aspects of physical and mental wellbeing. Protecting and promoting the mental wellbeing of the health and care workforce and trainees will be essential for the future of healthcare services post–pandemic.

One approach to achieving this is the digital package explored in this study. Globally, this was the first digital intervention to support the psychological wellbeing of health and care workers during the COVID-19 pandemic, developed in the UK, in March 2020 [21]. It is highly accessed - in the first 12 months, there were over 66,820 package users, worldwide—and it is deemed to be appropriate, meaningful and useful for the needs of health and care workers [31]. The focus of the digital package is on protecting and promoting mental wellbeing through raising awareness of the mental health impacts of COVID‐19, providing education around positive strategies and signposting to supportive resources. 

Subjective well–being (SWB) refers to how people experience and evaluate their lives and specific domains and activities in their lives [64]. The construct of mental wellbeing is complex and covers both affect and psychological functioning, with two distinct perspectives: the hedonic perspective, which focuses on the subjective experience of happiness and life satisfaction, and the eudaimonic perspective, focusing on psychological functioning and self realisation [37,65]. Elements of the digital package content draw on the core principles of positive psychology. Positive psychology is the scientific study of the factors that enable individuals (e.g., health and care workers, health and care trainees) and communities (e.g., healthcare workforce, healthcare organisations or educational institutions) to flourish. Positive mental wellbeing has major consequences for health and social outcomes [66,67], but is under–researched [37]. Our survey findings identified low mental wellbeing in healthcare trainees, as measured on a scale focusing entirely on the positive aspects of mental health (WEMWBS) [37]. This, coupled with our qualitive findings from the same sample, demonstrates a clear need to support and promote wellbeing in healthcare trainees. The package explored in this study provides a wealth of support and advice to facilitate engagement and motivation, build resilience and foster self–compassion—all factors that have been associated with mental wellbeing in students from the caring professions [68]. 

The pandemic was associated with significant disruption to studies, and major changes in ways of working. Prior studies have shown that delays in academic activities are related to aspects of mental health, such as increased anxiety [14]. Our qualitative findings showed that experiences of trainees varied dramatically, with some experiencing isolation due to fully remote working, and others rapidly deployed to support the healthcare services somewhat earlier than planned. All trainees experienced some level of disruption to academic studies and assessments, and/or opportunities for clinical learning, and for some this generated significant worry. Perceived access to academic and welfare support, the quality of relationship with (and regularity of contact from) academic tutors, and the transparency and timeliness of communication were highly variable across disciplines and institutions, but were uniformly perceived to be critical to mitigating the impact of the pandemic on healthcare trainees. While healthcare trainees are at risk of psychological distress, conversely, the disruption to learning and deployment into clinical roles for some, may also have provided self–efficacy building opportunities to identify new goals and approaches to facing the unknown [69], contribute to the global emergency effort, and ‘become stakeholders in the expansion and delivery of healthcare’ [70]. Some trainees have experienced a challenge of dual identity during this time; being part of the healthcare systems’ response to COVID-19 as a future healthcare professional, while at the same time perceived to be non–essential in clinical delivery [71] and occupying the position of learner in both the university and health sectors [72]. Importantly, many of our healthcare trainees alluded to imposter syndrome, and reported feeling trepidatious and unprepared for future clinical practice. It is essential that healthcare employers recognise the potential impacts of the COVID-19 pandemic on trainees’ confidence and preparedness for practice, as well as the mental health impacts of the pandemic which are prevalent and likely to be long–lasting.

The current study demonstrates that healthcare trainees perceive the digital package explored in this study to be a useful tool to augment the longer–term provision of psychological support for trainees and healthcare professionals alike, that will be useful during and after the pandemic. We found the package to have high fidelity (in terms of delivery and engagement) and excellent implementation qualities (in terms of practicality, resource challenges, attitudes, acceptability and usability) with a healthcare trainee sample. Qualitative findings showed that healthcare trainees found value in this resource, in terms of it raising awareness about mental wellbeing, validating mental health concerns in healthcare professions, and providing new knowledge and resources for personal use or signposting for their peers. This study demonstrates the potential for this resource to change attitudes in healthcare trainees, by enhancing their understanding of the impacts of a pandemic on other professional groups and across levels of seniority (thus promoting interprofessional learning and team cohesiveness), increasing sense of cultural humility (identified as important in client–facing professions [73]) and engendering positive behaviour change intentions with relation to identifying and managing signs of mental ill–health. 

We propose that this digital support package is widely distributed to healthcare trainees, but should be accompanied by details of local, institutional support for academic and welfare concerns. Regular check–ins, and wellbeing interventions will be essential to support the next generation health and care workforce, both in higher education and clinical settings. Trainees in our study suggested regular individual or small group check–ins from their institutions and future employers, to discuss educational or welfare concerns. They advocate for the provision of scheduled wellbeing activities to reduce isolation and alleviate stress, anxiety and low mood. Well–managed interventions should be put in place to ensure that qualifying trainees enter the health and care workforce adequately supported and mentored. 

## 5. Conclusions

Healthcare trainees are experiencing significant psychological impacts of COVID‐19, primarily related to risk of COVID-19 transmission, concerns about personal circumstances, and the longer–term impacts of disruption to studies during the pandemic and preparedness for future clinical roles. Negative culture within certain disciplines appears to hamper help–seeking around mental health, and provision of high–quality pastoral support is variable. Action should be taken to encourage open conversation about mental wellbeing. Wellbeing interventions will be essential to support the next generation health and care workforce, both in higher education and clinical settings. We found that an existing digital intervention that was developed to support wellbeing in health and care professionals is appropriate for healthcare trainees, with high fidelity and excellent implementation qualities. It is perceived to be a useful tool to augment longer–term provision of psychological support for healthcare trainees and professionals, during and after the COVID-19 pandemic.

## Figures and Tables

**Figure 1 ijerph-18-10647-f001:**
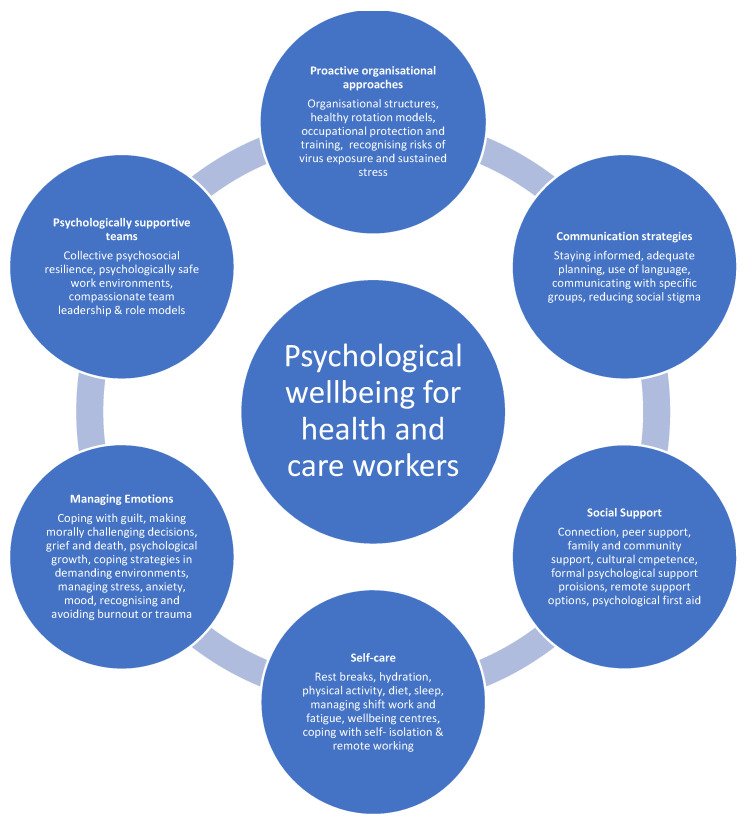
Conceptual model for mitigating the impacts of COVID-19 on health and care workers.

**Figure 2 ijerph-18-10647-f002:**
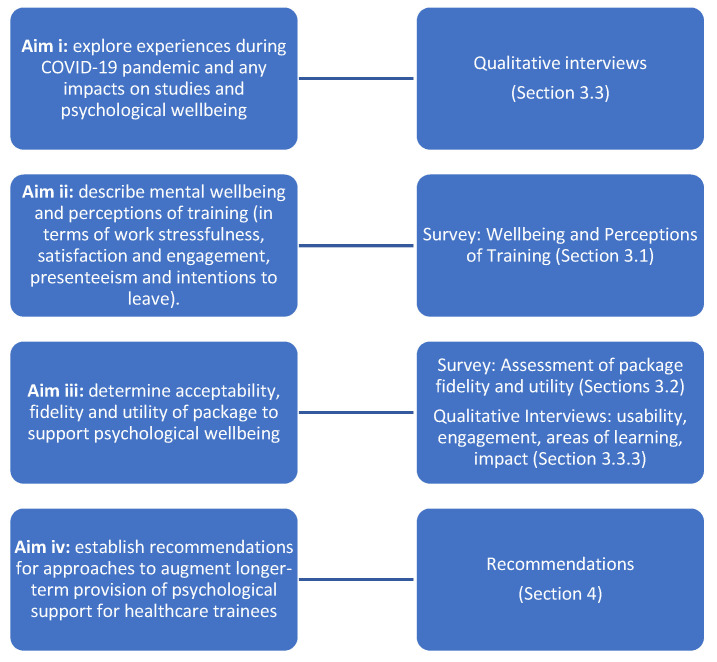
Approaches to data collection.

**Table 1 ijerph-18-10647-t001:** Participant Characteristics by Clinical Exposure.

Characteristics (*n* = 42, 100%)	Clinical Exposure ^+^ (*n* = 20, 47.6%)	No Clinical Exposure ^+^ (*n* = 22, 52.4%)	Full Sample (*n* = 42, 100%)
Age 16–20 21–30 31–40 41–50	4 (9.5) 8 (19) 7 (16.7) 1 (2.4)	9 (21.4) 10 (23.8) 3 (7.1) –	13 (31) 18 (42.9) 10 (23.8) 1 (2.4)
Gender Male Female Prefer not to disclose	6 (14.3) 13 (31) 1 (2.4)	3 (7.1) 19 (45.2) –	9 (21.4) 32 (76.2) 1 (2.4)
Ethnicity White Mixed Ethnicity BlackAfrican/Carribean/Black British Asian/Asian British Other ethnic group	14 (33.3) – 1 (2.4) 5 (11.9) –	9 (21.4) 1 (2.4) – 10 (23.8) 2 (4.8)	23 (54.8) 1 (2.4) 1 (2.4) 15 (35.7) 2 (4.8)
Year of Study 1 2 3 4 5 6+	3 (7.1) 7 (16.7 7 (16.7) 1 (2.4) 2 (4.8) –	2 (4.8) 12 (28.6) 5 (11.9) 1 (2.4) 1 (2.4) 1 (2.4)	5 (11.9) 19 (45.2) 12 (28.6) 2 (4.8) 3 (7.1) 1 (2.4)

Note: White–British: White, White–Irish, White—any other White background; Mixed: White and Asian, Any other mixed background, White and Black Caribbean; White and Black African; Asian/Asian British: Indian, Pakistani, Bangladeshi, any other Asian background; Black/Black British: Caribbean, African, any other Black background. ^+^ Clinical exposure refers to having been working in clinical settings at any point during the pandemic (placement or work).

**Table 2 ijerph-18-10647-t002:** Wellbeing and perceptions of training by clinical exposure (*n* = 42).

Item	Clinical Exposure Mean (s.d.) or *n* (%)	No Clinical Exposure Mean (s.d.) or *n* (%)	Comparison (p)
Course/training stressfulness	3.23 (0.92)	3.24 (0.73)	0.96
Course/training satisfaction	3.46 (1.12)	3.90 (0.81)	0.16
Intentions to leave	–	2 (4.8)	–
Presenteeism			0.18^a^
No, never	4 (9.5)	17 (40.5)	
Yes, once	3 (7.1)	9 (21.4)	
Yes, 2 to 5 times	5 (11.9)	3 (7.1)	
Yes, more than 5 times	1 (2.4)	0	
Work engagement			
Enthusiastic about training	5.08 (1.11) ^^^^	4.38 (0.97) ^^^	0.04 *
Training inspires me	4.69 (0.94) ^^^	4.17 (1.22) ^^^	0.18
Proud of my work	4.92 (1.24) ^^^^	4.38 (1.47) ^^^	0.27
Total UWES ^+^	4.76 (0.2) ^^^^	4.31 (0.19) ^^^	0.46 ^b^
WEMWBS Total	34.15 (8.13)	36.97 (9)	0.34 ^c^

* Significant at 0.05 alpha level; WEMWBS Warwick–Edinburgh Mental Wellbeing Scale; ^+^ UWES Utrecht Work Engagement Scale, dedication sub–scale ^^^ average (2.91–4.70), ^^^^ high (4.71–5.69); ^a^ Pearson correlation co–efficient; ^b^ Fisher–Freeman–Halton Exact Test; ^c^ Independent samples *t*-test.

**Table 3 ijerph-18-10647-t003:** Intervention fidelity and implementation testing.

Assessment Type (*n* = 42)	N	Actual Success Rate	Pre–Defined Success Rate
Fidelity Assessment		N (%) or mean (SD)	N (%) or mean
***Fidelity of Delivery*** Per–protocol delivery (functioning link) ***Toolkit completion rate:*** Main sections Further resources	42 42 42	41 (97.6) 41 (97.6) 19 (45.2)	>90% * >75% * >30% *
***Fidelity of Engagement*** Understanding of the toolkit Intervention receipt (perceived knowledge) Intervention enactment (knowledge use ^a^) Perceived enactment (future use ^b^)	42 42 19 23	41 (97.6) 33 (78.6) 19 (45.2) 20 (86.0)	>90% * >90% >30% * >50% *
Implementation Qualities		N (%) or mean (SD)	N (%) or mean
***Practicality*** Use by any healthcare professional Relevance to any healthcare professional Level of burden	42 42 42	40 (95.2) 7.24 (2.28) 5.55 (2.24)	>75% * >6 * <6 *
***Resource Challenges*** Time challenges Technical challenges (skills) Financial challenges	42 42 42	13 (31) 1 (2.4) 0 (0)	<25%* <25% * <25% *
***Attitudes*** Perceptions toward availability Would recommend to others	42 42	8.9 (1.41) 37 (88.1)	>6 * >75% *
***Acceptability*** Appropriate for needs Contains meaningful information Perceived usefulness of the toolkit	42 42 42	34 (81) 39 (92.9) 7.9 (1.96)	>75% * >75% * >6 *
***Usability*** Ease of navigation Technical difficulties (functioning)	42 42	8.5 (1.83) 3 (7.1)	>6 * <25% *
***Cost*** Acceptable cost implications	42	42 (100)	>75% *

^a^ Immediate actions following access to the package. ^b^ Item relates to hypothetical future action (intention) and is completed only those who had not enacted immediately. Note: * Meets pre–defined success rate.

**Table 4 ijerph-18-10647-t004:** Analytic Framework.

Overarching Themes	Sub–Themes	Codes
Impact of COVID-19 on studies	Level of exposure to COVID‐19	Clinical work in high–risk areas Placement exposures Contacts with family, partners or peers
	Impact of COVID-19 on healthcare studies	NHS deployment Remote/online working Changes in timetabling Student experiences (clinical practice, workload, isolation, work–life balance) Communication and information Quality of academic support
Emotional impacts of COVID-19	Emotional highs of the pandemic	Positive emotions Dealing with a crisis situation Knowledge and support
	Emotional lows of the pandemic	Negative emotions Concerns for the future Lack of support
	Discipline–specific impacts	Role responsibilities Expectations of the profession
	Ethnicity–specific impacts	Perceived vulnerability Risk and inequity
	Return to the ‘new normal’	Preparedness for practice Returning to study and placements Mental health validation and support
Digital support for psychological wellbeing (e–package †)	Usability and engagement	Accessibility and use Functionality and technology Comprehensiveness
	Areas of learning and impact	Most useful content New knowledge and learning
	Areas for future development	Least useful aspects Future support required
	Application of knowledge and learning	Personal value (new knowledge) Signposting resources (helplines, apps, links, videos) Attitude change (cultural competence, leadership) Behaviour change intentions

† Mitigating the psychological impact of COVID-19 on health and care workers [31].

## Data Availability

The data presented in this study are available on reasonable request from the corresponding author.

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
