# Peer review of "Psychological Impacts of COVID-19 on Healthcare Trainees and Perceptions towards a Digital Wellbeing Support Package"

_ijerph, 2021, doi:10.3390/ijerph182010647_

Round 1

Reviewer 1 Report

This paper contains a very interesting analysis. Revision is required for the following points.
First, it is necessary to focus on the thesis topic. It is necessary to present a research framework that can show the linkage between 3.1, 3.2, and 3.3 in which analysis is being performed.
Second, a theoretical discussion on subjective well-being should be added. In particular, it is necessary to review previous studies on variables affecting subjective well-being from a causal point of view.
Third, it is necessary to review what the theoretical meaning of the digital support package, which is being presented as an intervention tool, is.

Author Response

Reviewer 1

This paper contains a very interesting analysis.

Thank you for this positive feedback.

First, it is necessary to focus on the thesis topic. It is necessary to present a research framework that can show the linkage between 3.1, 3.2, and 3.3 in which analysis is being performed.

We have clarified the aims of the study to ensure that all analyses are appropriately reflected by aims i-iv:

The aims of the research were to:

i) explore the experiences of healthcare trainees during the COVID-19 pandemic and any impacts on their studies and psychological wellbeing,

ii) describe trainees’ mental wellbeing and perceptions of training (in terms of work stressfulness, satisfaction and engagement, presenteeism and intentions to leave).

iii) determine the acceptability, fidelity and utility of an e-learning package to support psychological wellbeing in healthcare trainees,

iv) establish recommendations for approaches to augment longer-term provision of psychological support for healthcare trainees, during and after the COVID-19 pandemic.”

We have included a new figure to map data collection approaches to aims, and it is now clearer which results sections relate to the four specific aims of the study through this added text:

“Data collection approaches are mapped to study aims and corresponding results sections (Figure 2). Data were collected by qualitative interviews and a structured survey was completed by all interview participants (to meet aims i-iii). Findings are synthesised in a discussion with recommendations (to meet aim iv).”

Second, a theoretical discussion on subjective well-being should be added. In particular, it is necessary to review previous studies on variables affecting subjective well-being from a causal point of view.

Text has been added to the discussion to demonstrate the use of wellbeing in this context, including a definition of subjective wellbeing, the link with positive psychology, and reference to a study of factors associated with wellbeing in the caring professions. We have endeavoured to address this comment while mindful of the length of the manuscript and the need to remain focused on reporting the data we have.

Third, it is necessary to review what the theoretical meaning of the digital support package, which is being presented as an intervention tool, is.

We have included the conceptual model for the digital support package (new Figure 1). We have added text to the introduction and discussion to further demonstrate the theoretical link between the package content and wellbeing theory / model.

General revisions

We have added the following sections:

Funding: The study was supported by the University of Nottingham INSPIRE Academic Medicine Society Summer Research Internship Programme, as a rapid response to the COVID-19 pandemic. INSPIRE is a nationwide programme supported by the Wellcome Trust to engage medical students in research.

Institutional Review Board Statement: The study was conducted according to the guidelines of the Declaration of Helsinki and was approved by the University of Nottingham Faculty of Medicine and Health Sciences Research Ethics Committee on 11th June 2021 (FMHS REC 39-0620).

Informed Consent Statement: Informed consent was obtained from all participants involved in the study.

Data Availability Statement: The data presented in this study are available on request from the corresponding author. The data are not publicly available due to risk of participant identification.

Acknowledgments: The authors would like to thank Kalp Patel for initial support with study promotion, and the organisers of the INSPIRE Academic Medicine Society Summer Research Internship Programme.

We have added a sentence to the discussion to emphasise diversity of views gathered.

We have corrected some minor grammatical and typographical errors.

We have clarified the exact date of ethical approval in the methods and Institutional Review Board Statement.

Following a check on the number of users of the digital package after 12 months, we have replaced the previous estimate of 80,000 with ‘over 66,820 package users’

To further demonstrate the rigour of the study, we have added statements:

“All data were collected by independent researchers who had no involvement in the design or development of the digital package.”

“Survey data were analysed by an independent researcher (MY) who had no involvement in recruitment, intervention, data collection or analysis of qualitative data”.

We added a space between sub-themes so the text is easier to read.

The acronym for the study has been included in the abstract and methods (in order that the paper is clearly aligned with the study registration on clinicaltrials.gov)

Reviewer 2 Report

I have read with great interest the present manuscript on the psychological impacts of the pandemic on HC trainees and the role of an online package to support psychological wellbeing of trainees. Althought the study was performed with few participants (only 42 trainees), the study is very interesting and should be considered for being accepted. The study is well defined, well written.

Author Response

I have read with great interest the present manuscript on the psychological impacts of the pandemic on HC trainees and the role of an online package to support psychological wellbeing of trainees. Although the study was performed with few participants (only 42 trainees), the study is very interesting and should be considered for being accepted. The study is well defined, well written.

Thank you for this positive feedback.

General revisions

We have added the following sections:

Funding: The study was supported by the University of Nottingham INSPIRE Academic Medicine Society Summer Research Internship Programme, as a rapid response to the COVID-19 pandemic. INSPIRE is a nationwide programme supported by the Wellcome Trust to engage medical students in research.

Institutional Review Board Statement: The study was conducted according to the guidelines of the Declaration of Helsinki and was approved by the University of Nottingham Faculty of Medicine and Health Sciences Research Ethics Committee on 11th June 2021 (FMHS REC 39-0620).

Informed Consent Statement: Informed consent was obtained from all participants involved in the study.

Data Availability Statement: The data presented in this study are available on request from the corresponding author. The data are not publicly available due to risk of participant identification.

Acknowledgments: The authors would like to thank Kalp Patel for initial support with study promotion, and the organisers of the INSPIRE Academic Medicine Society Summer Research Internship Programme.

We have added a sentence to the discussion to emphasise diversity of views gathered.

We have corrected some minor grammatical and typographical errors.

We have clarified the exact date of ethical approval in the methods and Institutional Review Board Statement.

Following a check on the number of users of the digital package after 12 months, we have replaced the previous estimate of 80,000 with ‘over 66,820 package users’

To further demonstrate the rigour of the study, we have added statements:

“All data were collected by independent researchers who had no involvement in the design or development of the digital package.”

“Survey data were analysed by an independent researcher (MY) who had no involvement in recruitment, intervention, data collection or analysis of qualitative data”.

We added a space between sub-themes so the text is easier to read.

The acronym for the study has been included in the abstract and methods (in order that the paper is clearly aligned with the study registration on clinicaltrials.gov)

Reviewer 3 Report

In the manuscript “Psychological impacts of COVID-19 on healthcare trainees and perceptions towards a digital wellbeing support package”, the authors, conducting a mixed-methods study, explore the experiences of healthcare trainees during the COVID-19 pandemic and the impact on studies and psychological well-being, and analyze the acceptability, fidelity and utility of an e-learning package to support psychological well-being in healthcare trainees. The manuscript deals with relevant and current topics; however, some minor changes should be done before publication:

  1. On page 3, lines 96 to 99 say the following: “Eligible participants were health and medical trainees registered for study during the COVID-19 pandemic, purposively selected to represent diversity across higher education institution, gender and discipline of study (including medicine, nursing, and allied health). Participants were recruited from 13 universities in the UK…”

But, as stated in the abstract, on page 1 line 14, the participants were 9 males and 33 females. It is not clear why if a sample to represent diversity across gender was desired, only 9 males were included in the study.

  1. "Table 2. Wellbeing and perceptions of training by clinical exposure (n=42)" (on page 6), should be reviewed as it is confusing or may contain errors. Specifically, when comparing the wellbeing between Clinical exposure and No clinical exposure participants in Presenteeism, categorized into 4 groups (No, never; Yes, once; Yes, 2 to 5 times; Yes, more than 5 times), in the Comparison (p) column appears “.18a” And, according to the footnote “a” is the Pearson Correlation Coefficient. It is not understood between which values such a Pearson correlation coefficient has been obtained.

Author Response

On page 3, lines 96 to 99 say the following: “Eligible participants were health and medical trainees registered for study during the COVID-19 pandemic, purposively selected to represent diversity across higher education institution, gender and discipline of study (including medicine, nursing, and allied health). Participants were recruited from 13 universities in the UK…”

But, as stated in the abstract, on page 1 line 14, the participants were 9 males and 33 females. It is not clear why if a sample to represent diversity across gender was desired, only 9 males were included in the study.

Thank you for this observation – we agree this needed to be clearer. Female participants were purposely over-sampled. This was to reflect a higher proportion of women in the UK healthcare workforce overall (NHS employees: 77% female NHS Employers, 2019), and the gender balance in healthcare education (nursing students: 90% female; allied health students: 75% female Office for Students, 2020; medicine and dentistry students: 64% UCAS, 2020).

We have added this to the text in the methods and provided three supporting references.

NHS Employers. Gender in the NHS infographic. 12 May 2019. Available at: https://www.nhsemployers.org/articles/gender-nhs-infographic (accessed 03.10.21)

Office for Students. Male participation in nursing and allied health higher education courses. 22 January 2020. Available at: https://www.officeforstudents.org.uk/publications/male-participation-in-nursing-and-allied-health-higher-education-courses/  (accessed 03.10.21).

Universities and College Admissions Service. Undergraduate Statistics and Reports, 2020. Available at: https://www.ucas.com/undergraduate-statistics-and-reports  (accessed 03.10.21)

"Table 2. Wellbeing and perceptions of training by clinical exposure (n=42)" (on page 6), should be reviewed as it is confusing or may contain errors. Specifically, when comparing the wellbeing between Clinical exposure and No clinical exposure participants in Presenteeism, categorized into 4 groups (No, never; Yes, once; Yes, 2 to 5 times; Yes, more than 5 times), in the Comparison (p) column appears “.18a” And, according to the footnote “a” is the Pearson Correlation Coefficient. It is not understood between which values such a Pearson correlation coefficient has been obtained.

This number (.18a) presents the presenteeism comparison between clinical exposure and no clinical exposure including all answers (no, yes, 2-5 and 5+). The coefficient was obtained from all values. The problem was that the line (.18a) shifted to midline during the editorial processes, this has been corrected. All other Tables have been checked and are correct.

General revisions

We have added the following sections:

Funding: The study was supported by the University of Nottingham INSPIRE Academic Medicine Society Summer Research Internship Programme, as a rapid response to the COVID-19 pandemic. INSPIRE is a nationwide programme supported by the Wellcome Trust to engage medical students in research.

Institutional Review Board Statement: The study was conducted according to the guidelines of the Declaration of Helsinki and was approved by the University of Nottingham Faculty of Medicine and Health Sciences Research Ethics Committee on 11th June 2021 (FMHS REC 39-0620).

Informed Consent Statement: Informed consent was obtained from all participants involved in the study.

Data Availability Statement: The data presented in this study are available on request from the corresponding author. The data are not publicly available due to risk of participant identification.

Acknowledgments: The authors would like to thank Kalp Patel for initial support with study promotion, and the organisers of the INSPIRE Academic Medicine Society Summer Research Internship Programme.

We have added a sentence to the discussion to emphasise diversity of views gathered.

We have corrected some minor grammatical and typographical errors.

We have clarified the exact date of ethical approval in the methods and Institutional Review Board Statement.

Following a check on the number of users of the digital package after 12 months, we have replaced the previous estimate of 80,000 with ‘over 66,820 package users’

To further demonstrate the rigour of the study, we have added statements:

“All data were collected by independent researchers who had no involvement in the design or development of the digital package.”

“Survey data were analysed by an independent researcher (MY) who had no involvement in recruitment, intervention, data collection or analysis of qualitative data”.

We added a space between sub-themes so the text is easier to read.

The acronym for the study has been included in the abstract and methods (in order that the paper is clearly aligned with the study registration on clinicaltrials.gov)